# Genome-Wide Small RNA Sequencing Identifies MicroRNAs Deregulated in Non-Small Cell Lung Carcinoma Harboring Gain-of-Function Mutant p53

**DOI:** 10.3390/genes10110852

**Published:** 2019-10-28

**Authors:** Arindam Datta, Pijush Das, Sanjib Dey, Sangeeta Ghuwalewala, Dishari Ghatak, Sk. Kayum Alam, Raghunath Chatterjee, Susanta Roychoudhury

**Affiliations:** 1Cancer Biology and Inflammatory Disorder Division, CSIR-Indian Institute of Chemical Biology, 4, Raja S.C. Mullick Road, Jadavpur, Kolkata 700032, India; hereisarindam@gmail.com (A.D.); topijush@gmail.com (P.D.); sanby19@gmail.com (S.D.); sangeetaghuwalewala@gmail.com (S.G.); dishari18@gmail.com (D.G.); kayumalam86@gmail.com (S.K.A.); 2Human Genetics Unit, Indian Statistical Institute, 203 B. T. Road, Kolkata 700108, India; 3Saroj Gupta Cancer Centre and Research Institute, Mahatma Gandhi Road, Thakurpukur, Kolkata 700063, India

**Keywords:** microRNA, non-small cell lung carcinoma, mutant p53, gain of function, small RNA sequencing, TCGA, novel miRNA, node metastasis, survival

## Abstract

Mutations in the *TP53* gene are one of the most frequent events in cancers. Certain missense mutant p53 proteins gain oncogenic functions (gain-of-functions) and drive tumorigenesis. Apart from the coding genes, a few non-coding microRNAs (miRNAs) are implicated in mediating mutant p53-driven cancer phenotypes. Here, we identified miRNAs in mutant p53^R273H^ bearing non-small cell lung carcinoma (NSCLC) cells while using small RNA deep sequencing. Differentially regulated miRNAs were validated in the TCGA lung adenocarcinoma patients with p53 mutations and, subsequently, we identified specific miRNA signatures that are associated with lymph node metastasis and poor survival of the patients. Pathway analyses with integrated miRNA-mRNA expressions further revealed potential regulatory molecular networks in mutant p53 cancer cells. A possible contribution of putative mutant p53-regulated miRNAs in epithelial-to-mesenchymal transition (EMT) is also predicted. Most importantly, we identified a novel miRNA from the unmapped sequencing reads through a systematic computational approach. The newly identified miRNA promotes proliferation, colony-forming ability, and migration of NSCLC cells. Overall, the present study provides an altered miRNA expression profile that might be useful in biomarker discovery for non-small cell lung cancers with *TP53* mutations and discovers a hitherto unknown miRNA with oncogenic potential.

## 1. Introduction

Mutations in the tumor suppressor *TP53* gene influence cancer progression and clinical outcomes of human cancers [1,2]. Most of these mutations are monoallelic missense point mutations that result in the synthesis of full length mutant p53 proteins with altered functions [3]. These mutations generally occur at high frequencies in six “hot spot” amino acid residues [3] of the central DNA-binding domain of p53, which causes a loss of its sequence-specific DNA-binding activity. In addition to the loss of wild-type tumor suppressor properties, mutant p53 gains new functions (GOFs) to promote various oncogenic phenotypes, including cancer cell proliferation, increased DNA replication, genomic instability, invasion, metastasis, and increased chemo-resistance [4,5,6,7,8]. GOF mutant p53, designated as an oncogenic transcription factor, can modulate the expression of several genes that are involved in oncogenic processes [9]. By cooperating with other transcription factors, such as NF-Y and Sp1, mutant p53 is recruited to target promoters and it facilitates the transcription of the respective genes [9]. Physical interactions of mutant p53 with tumor suppressors p63 and p73 sequesters these proteins and inhibits the transactivation of their respective target genes [5,10,11]. Moreover, in response to DNA damage, GOF mutant p53 transactivates cellular genes by recruiting histone modifiers [12].

Lung cancer is one of the leading causes of cancer-related deaths across the world [13]. Approximately 80% of all primary lung cancer cases are classified as non-small cell lung cancer (NSCLC) [14,15] and more than 50% of NSCLC patients generally carry *TP53* mutations that predict poor prognosis [14,15,16]. These findings suggest that *TP53* mutation determines malignant progression in NSCLC. Among the six “hot spot” missense point mutations, R273 is one of the most frequently mutated (6.7%) residues in human cancers [12], particularly in the NSCLC (~5%) (IARC database, http://www-p53.iarc.fr). Mutant p53^R273H^ has been reported to confer enhanced chemo-resistance and increased cell migration in NSCLC cell line H1299 [17,18]. Moreover, several in vitro and in vivo studies demonstrated the ability of this p53 mutant to induce GOF properties, such as cancer cell invasion, survival, and proliferation; increased migration; drug resistance; anchorage-independent growth; and, genomic instability [19].

The pivotal role of miRNAs in human cancer is well established. Several oncogenic and tumor-suppressive miRNAs have already been identified [20]. The role of miRNAs in mediating tumor suppressor functions of wild-type p53 is also well documented [21]. Genome-wide studies have identified wild-type p53-regulated miRNAs that contribute to tumor suppression and stress responses [22,23]. Although the link between wild type p53 and miRNA is well established, the role of mutant p53 in regulating cellular miRNAs is still emerging. Donzelli et al. first reported that miR-128b is transcriptionally regulated by mutant p53 and it confers chemo-resistance to lung cancer cells [24]. Another report demonstrated that, upon DNA damage, down-regulation of miR-223 by GOF p53^R175H^ via ZEB-1 (a transcriptional repressor) contributes to chemo-resistance of cultured tumor cells [25]. A few other cellular miRNAs (e.g., let-7i, miR-130b, -27a, and -155) is also implicated in mutant p53-driven cancer cell invasion, metastasis, epithelial-to-mesenchymal transition (EMT), and proliferation [26,27,28,29]. These evidences suggest that miRNA is a critical mediator of mutant p53 GOF properties in cancer cells. Therefore, identification of mutant p53-regulated miRNAs on a genome-wide scale is of paramount importance in mutant p53 gain-of-function research. 

In this study, using small RNA deep sequencing, we identified GOF mutant p53^R273H^ regulated miRNAs in NSCLC cells. We subsequently explored mutant p53^R273H^-regulated miRNA-mRNA molecular networks and their functions through systematic computational analyses. miRNAs that were obtained from our sequencing experiment were further validated in The Cancer Genome Atlas (TCGA) lung adenocarcinoma patient dataset. Moreover, our analyses identified mutant p53-regulated miRNAs that are associated with lymph node metastasis and poor clinical outcome in lung cancer patients. Most importantly, we discovered a novel miRNA that appeared to be involved in augmenting oncogenic phenotypes in cancer cells. Collectively, the findings of the present study further enrich the evolving knowledge of GOF mutant p53-regulated miRNAs and present a potential novel miRNA with oncogenic functions. 

## 2. Materials and Methods

### 2.1. Cell Culture 

To generate mutant p53^R273H^ expressing stable H1299 cell line, the cells were first transfected with pCMV-p53R273H expression plasmid (pCMV-Neo-Bam-p53 R273H, kindly provided by Bert Vogelstein, Johns Hopkins Kimmel Cancer Center, Baltimore, MD, USA) while using Lipofectamine 2000 (Invitrogen, Thermo Fisher Scientific Inc., Waltham, MA, USA). Forty-eight hours post-transfection, the cells were sub-cultured at a ratio of 1:6 density in media supplemented with G418 (Life Technologies, Thermo Fisher Scientific Inc., Waltham, MA, USA) at a final concentration of 800 µg/mL. After 15–20 days, G418 resistant colonies were selected and propagated in G418 containing (400 µg/mL) medium to generate mutant p53^R273H^ expressing stable cell lines. H1299 cells that were infected with an empty vector were kindly provided by Varda Rotter (Weizmann Institute of Science, Rehovot, Israel). A549 cell line was purchased from ATCC. All cell lines were cultured in RPMI 1640 medium (Life Technologies, Thermo Fisher Scientific Inc., Waltham, MA, USA) supplemented with 10% fetal calf serum, 1% Pen-Strep, and 0.006% Gentamicin (Life Technologies, Thermo Fisher Scientific Inc., Waltham, MA, USA). STR profiling confirmed the isogeneity of H1299/EV and H1299/ mutant p53^R273H^ cell lines, as described previously [6].

### 2.2. Western Blotting

Cells that were grown in 35 mm dish were lysed in NP-40 cell lysis buffer (Life Technologies, Thermo Fisher Scientific Inc., Waltham, MA, USA) supplemented with protease inhibitor cocktail (Sigma Aldrich, St. Louis, USA) and 30 µg of total protein was immunoblotted with antibodies against p53 (DO-1 (SC-126), Santa Cruz Biotechnology, CA, USA), and β-actin (# A5316, Sigma Aldrich, St. Louis, MO, USA).

### 2.3. RNA Isolation, Small RNA Library Preparation, and Deep Sequencing

Small RNA fractions from cells were isolated while using PureLink miRNA Isolation kit (Ambion, Austin, TX, USA) according to the manufacturer’s protocol. Briefly, in a two-column based purification method, the cells were first resuspended in binding buffer containing guanidine isothiocyanate and then mixed with ethanol (final concentration 35%). It was then passed through a spin column and the flow through containing the small RNAs was collected. Ethanol (final concentration 70%) was added to the flow, followed by extraction though silica-based columns, where the small RNAs bind to the silica membrane. The impurities were subsequently removed by washing the columns with wash buffer and small RNAs were eluted in RNase free water. The enrichment of miRNAs in small RNA samples was assessed by Qubit ® 2.0 Fluorometer (Life Technologies, Thermo Fisher Scientific Inc., Waltham, MA, USA) and Agilent® 2100 Bioanalyzer using Small RNA kit (Agilent Technologies, Santa Clara, CA, USA) (Appendix A). Total RNA-Seq kit V2.0 (Life Technologies, Thermo Fisher Scientific Inc., Waltham, MA, USA) was used to prepare small RNA cDNA libraries. Briefly, small RNAs (~150–250 ng) were first mixed with Hybridization Solution and Ion Adaptor Mix v2, followed by hybridization in a thermal cycler. Ligation enzyme mix was added to the hybridization reactions and then incubated in a thermal cycler at 16 °C for 16 h. Adapter-ligated small RNA samples were subsequently reverse transcribed with Ion RT Primer v2 and SuperScript® III Enzyme Mix to make cDNAs. The cDNA products were purified and size-selected while using the magnetic bead clean-up module and PCR amplified with Ion 5′ and 3′ PCR Primers. The amplified DNAs were quantified and assessed for size distribution using Agilent® DNA 1000 Kit in Agilent® 2100 Bioanalyzer. The cDNA libraries were clonally amplified on Ion Sphere™ Particles (ISPs) by emulsion PCR in Ion OneTouch™ System using Ion OneTouch™ 200 Template Kit v2 DL (Life Technologies, Thermo Fisher Scientific Inc., Waltham, MA, USA). The enrichment of the template positive ion spheres was carried out in Ion OneTouch™ Enrichment System. Template enriched ion sphere particles were subsequently sequenced in Ion PGM™ sequencer (Model No. 508-U001, Life Technologies, Thermo Fisher Scientific Inc., Waltham, MA, USA) using Ion 316™ Chip and Ion PGM™ 200 Sequencing Kit (Life Technologies, Thermo Fisher Scientific Inc., Waltham, MA, USA). The Torrent Suite Software (Life Technologies, Thermo Fisher Scientific Inc., Waltham, MA, USA) analysis pipeline was used to process raw data that were acquired from the Ion PGM™ sequencer to generate read files containing high-quality bases and output base calls in FASTQ file formats.

Raw sequence reads, as FASTQ format, were pre-processed using FASTX-toolkit (http://hannonlab.cshl.edu/fastx_toolkit/) to generate high-quality reads. Reads were trimmed at 3′ end and the base quality value was calculated while using the Phred quality score (Q ≥ 20). We only considered sequencing reads of length between 17 bp to 35 bp (Appendix A). High-quality raw reads were aligned to the reference miRBase (miRBase20; http://www.mirbase.org/) while using SHRiMP aligner (http://compbio.cs.toronto.edu/shrimp/). After alignment, an in-house Perl script was used for counting the reads that were aligned to miRNA, tRNA, rRNA, and adapter sequences. Reads not aligned in SHRiMP aligner were used for further mapping to the human genome (GRCh37/hg19; https://genome.ucsc.edu/) while using Novoalign with miRNA mode (http://www.novocraft.com/). The miRNA read counts were adjusted according to the aligned reads from both SHRiMP and Novoalign. miRNAs with three or more reads were used for differential expression analysis in R package DESeq. The remaining reads that were mapped to the genome beyond the annotated miRNAs were further used for novel miRNA prediction. 

### 2.4. Quantitative Real Time PCR

Total RNA from cell lines was isolated using TRIZOL (Invitrogen, Thermo Fisher Scientific Inc., Waltham, MA, USA), according to the manufacturer’s protocol. Around 250 ng of total RNA was reverse transcribed to prepare cDNAs using either miRNA specific stem-loop primers [30] or miScript II RT Kit (Qiagen, Hilden, Germany). Real time PCR reactions of cDNAs were carried out with forward primers specific to each miRNA in 7500 Fast and StepOnePlus Real-Time PCR Systems (Applied Biosystems, Foster City, CA, USA) while using Fast start universal SYBR Green master mix (Roche, Penzberg, Germany). U6 snRNA was used as endogenous reference control. Fold change values (2^−ΔΔ*C*^_T_) were calculated from the mean of three independent experiments. Two-tailed student’s *t* test was used to compute statistical significance. Primer sequences for mature miRNAs are listed in Appendix A.

### 2.5. Prediction of Novel MiRNAs 

Quality processed reads that were not mapped to miRBase 20 using SHRiMP aligner were further used for novel miRNA prediction. Reads were aligned to the human genome assembly (GRCh37/hg19) while using a Novoalign aligner in miRNA mode to predict the potential miRNA hairpin loci in the genome. The identified loci were subsequently mapped to the miRNA precursor sequences present in miRBase 20. Reads that mapped to known miRNA loci were used to adjust the read count in SHRiMP predicted miRNAs. The remaining reads were used for subsequent analyses in novel miRNA prediction. Candidate sequences that had minimum four read counts across the samples and aligned to loci not reported in miRBase 20 were considered for further analyses. The predicted hairpin sequences were analyzed in mfold 3.2 for secondary structure prediction. Candidate sequences that showed characteristic precursor stem-loop structure with minimum free energy (MFE) ΔG ≤ −20 kcal/mol and formed the stem region of the mature miRNA were considered as potential novel miRNAs.

### 2.6. miRNA Target Prediction and Pathway Analysis in IPA

Differentially regulated miRNAs by mutant p53^R273H^ were analyzed while using the Ingenuity Pathway Analysis (IPA; https://analysis.ingenuity.com) tool to identify their target genes, pathways, and molecular networks. The selected miRNAs were uploaded in IPA and miRNA target filter was used to identify their predicted and experimentally validated targets. Significantly enriched cellular pathways, biological functions, and genetic networks by a given set of miRNAs were generated in IPA. Integrated miRNA-mRNA expression analyses of the differentially regulated miRNAs and target mRNAs in H1299/EV and H1299/mutant p53^R273H^ cells were carried out while using the expression pairing tool in IPA. Target genes of novel miRNA were predicted using miRanda [31]. The “seed” sequence of the novel miRNA (2–8 bases at the 5′ end) was aligned to the 3′ UTRs of transcripts downloaded from the UCSC GRCh37/hg19 for 100% complementarily with miRanda score ≥ 150 and MFE ≤ −20 kcal/mol. The predicted targets were further analyzed in IPA for pathway enrichment. 

### 2.7. Analysis of Publicly Available MiRNA Datasets

The TCGA miRNA sequencing dataset of lung adenocarcinoma patients was downloaded from the TCGA data portal. The dataset contained miRNA seq data (level 3) of 230 lung adenocarcinoma patients that were sequenced on Illumina GA and Illumina Hiseq platforms (https://tcga-data.nci.nih.gov/docs/publications/luad_2014/) [32]. The clinical and genetic information of 230 patients were obtained from the article describing the study and cBioPortal (http://cbioportal.org) [32,33,34]. Lung adenocarcinoma patients were classified based on their *TP53* mutation status for validation of the mutant p53^R273H^-regulated miRNAs in TCGA dataset. The types of *TP53* mutation present in 230 patients were identified using *TP53* somatic mutation dataset in the IARC *TP53* Database [2] (version R17, November 2013). Patients harboring truncated p53 were not considered in our analysis because only p53 missense mutants are known to confer GOF properties [35]. Out of 47 differentially regulated miRNAs in H1299/mutant p53^R273H^ cells, we only considered 34 miRNAs that showed read coverage ≥ 2 in the TCGA dataset. Unsupervised hierarchical clustering of TCGA patient samples was done with complete linkage using Euclidean distance as a distance metrics based on their normalized miRNA read counts to remove the outliers. After outlier elimination, differential expression analysis of miRNAs was carried out between patients with wild-type and missense mutant p53 applying the empirical Bayes, moderated t-statistics in Bioconductor package LIMMA. 

Patients were divided into node-positive (N_+,_
*n* = 85) and node-negative (N_0,_
*n* = 138) groups based on their N stage status to identify the mutant p53^R273H^ regulated miRNAs that are involved in lymph node metastasis. Patients without N stage information (*n* = 7) were excluded from the analysis. The Mann–Whitney test was applied to determine the statistical significance of the difference between the relative expression of seven mutant p53-regulated miRNAs in N_+_ and N_0_ group of patients. Receiver operating characteristic (ROC) analyses were done in GraphPad Prism version 5.03. Survival analysis for seven mutant p53 regulated miRNAs was done using the Kaplan–Meier estimate in GraphPad Prism version 5.03. For survival analysis, we only considered those patients whose survival data was available in the TCGA clinical dataset (*n* = 195) obtained from the cBioPortal and were classified based on high (≥75th percentile) and low (≤25th percentile) expression of the respective miRNA. 

NCI-60 miRNA expression dataset (GEO accession number GSE26375) was analyzed to obtain the expression values of the mutant p53-regulated miRNAs. The NCI-60 cell lines were further classified as mesenchymal and epithelial cell types that are based on E-Cadherin/Vimentin expression ratio, as reported by Sun-Mi Park et al [36]. Subsequently, bioconductor package LIMMA was applied to compare the relative expression of these miRNAs between epithelial and mesenchymal groups of NCI-60 cell lines [36].

Small RNA sequencing data of 20 lung adenocarcinoma tissues and 10 squamous cell carcinoma, and 30 corresponding paired noncancerous lung tissues from the same patients were downloaded from the GEO database (GSE33858). Read alignment files (BAM) for these samples were used to identify the expression profile of newly identified miRNA. htseq-count within the HTSeq Python library were used to determine the count of known as well as newly identified miRNAs from both tumor and corresponding noncancerous tissues [37]. Differential expression analysis of miRNAs was performed while using DESeq [38]. 

### 2.8. Cloning of Novel MiRNA

The newly identified miRNA (miR-X) precursor sequence was amplified from genomic DNA using specific primers having KpnI and HindIII restriction enzyme sites at 5′ and 3′ ends, respectively. Following digestion with KpnI and HindIII, PCR amplified product was cloned into the pRNA-U6.1 miRNA expression vector. The ligated products were transformed in *E.coli* DH5α and positive clones were selected by restriction digestion and sequencing.

### 2.9. Cell Proliferation Assay

Cells that were transfected with pRNA-U6.1-miR-X or control empty vector were seeded at 4000 cells/well of 96 well plate and cultured in complete medium up to seven days. Cell proliferation was measured on individual days while using WST1 cell proliferation reagent (Roche, Penzberg, Germany) according to the manufacturer’s protocol. Relative viability index was calculated in MS-Excel from the average absorbance values that were obtained from three biological replicates.

### 2.10. Intracellular Ki-67 Staining 

Cells that were transfected with pRNA-U6.1 vector or pRNA-U6.1-miR-X plasmid were trypsinized and resuspended in wash buffer (1X PBS containing 1% FBS and 0.02% sodium azide) 48 h post-transfection. Cells were fixed in 1% paraformaldehyde for 15 min. and then washed with wash buffer followed by permeabilization with 1X Perm 2 solution (BD, Franklin Lakes, New Jersey, USA). The cells were washed with saponin wash buffer and resuspended in saponin wash buffer containing Ki-67 antibody (1:50, D2H10, Rabbit mAB, Cell signaling technology, Danvers, MA, USA) or isotype control (Rabbit mAb IgG (DA1E) XP^R^, Cell signaling technology) and incubated for 40 min at room temperature. The cells were washed and incubated with goat anti-rabbit Alexa Fluor 488 secondary antibody (1:100, Molecular Probes, Invitrogen, Carlsbad, CA, USA) for 30 min in the dark. The cells were washed three times with saponin wash buffer and finally resuspended in 1% paraformaldehyde solution. The stained cells were analyzed by FACS in BD LSRFortessa while using BD FACSDiva 6.2 software. 

### 2.11. Colony Formation Assay

Around 1500 cells were seeded on 6 cm dishes and allowed to grow in complete medium for 10–15 days until the appearance of visible colonies. The colonies were stained with 0.1% methylene blue for 30 min at room temperature, followed by washing with 1X PBS. Stained colonies were photographed and counted while using ImageJ software (http://rsb.info.nih.gov/ij/). The average number of colonies from three independent biological experiments were plotted in GraphPad Prism version 5.03.

### 2.12. Monolayer Wound Healing Assay

Approximately 1 × 10^6^ cells plated on a six-well plate were transfected with control pRNA-U6.1 vector or pRNA-U6.1-miR-X and they were grown for 30 h before the scratch was introduced. Photographs were taken at 0 h and 16–18 h after the scratch was introduced using a microscope (Olympus 1 × 51, camera Jenoptik). The relative distance that was migrated by the cells over time was measured in order to compare the wound healing capability of control and miR-X overexpressing cells. The percentage of average distance migrated by the cells was calculated from the data that were obtained from three independent experiments.

### 2.13. Availability of Data and Materials

Small RNA sequencing data have been submitted to the GEO database with the accession number GSE68353

## 3. Results

### 3.1. Small RNA Sequencing Identifies MiRNAs Deregulated in GOF Mutant p53 Cancer Cells

We carried out small RNA sequencing in NSCLC cell line H1299 either harboring a control empty vector (H1299/EV) or stably expressing mutant p53^R273H^ (H1299/mutant p53^R273H^) to identify GOF mutant p53-regulated miRNAs on a genome-wide scale (Figure 1A). Normalized read counts of each group showed a good correlation between biological replicates with an average Pearson correlation coefficient > 0.9 (Figure 1B, Table 1). Approximately 72 to 80% reads were mapped to the reference miRBase (miRBase20; http://www.mirbase.org/) in both H1299/EV and H1299/mutant p53^R273H^ cells (Table 1). Differential expression analysis of the mapped miRNAs yielded forty-seven differentially regulated miRNAs between the control and mutant p53^R273H^ expressing cells (*p*-value < 0.05; fold change ≥ 2) (Figure 1C). Among these, 21 miRNAs were found to be up-regulated and 26 were down-regulated in H1299/mutant p53^R273H^ cells. Subsequent qRT-PCR-based validation of seven randomly selected miRNAs in the control and mutant p53^R273H^ cells showed concordant results with the genome-wide sequencing data (Figure 1D). The differentially regulated miRNAs were subjected to pathway analysis while using the Ingenuity Pathway Analysis tool (IPA; https://analysis.ingenuity.com) to gain further insight into the biological changes that they could impart in mutant p53 cells. Crucial cellular functions, such as the regulation of cell cycle, cellular development, growth and proliferation, movement, and DNA replication, recombination, and repair were significantly enriched in our analyses (Table 2). Notably, most of the pathways enriched are generally found to be altered in cancer and affected by GOF mutant p53 [4,5]. Three miRNA-mRNA networks (IPA score ≥ 25) were significantly enriched by these miRNAs (Figure 2 and Appendix A). Among the significantly enriched networks, TP53 was found to either be targeted by, or regulate many, miRNAs that were deregulated in GOF mutant p53 cells (Figure 2). Downregulated miRNAs, like mir-194 and mir-296, were the targets of p53 and possibly works in a positive feedback loop with wildtype p53, while upregulated miRNAs, mir-132, mir-140, and mir-17 were either directly or indirectly regulated by mutant p53. Apart from *TP53*, ephrin-B family receptor *EPHB6*, transforming growth factor β (*TGFBI*), and Insulin were also enriched as major focal nodes in the miRNA-mRNA networks (Figure 2). Interactions with EphB6, TGFβ1, and Insulin suggest the possible contribution of these crucial signaling factors in mutant p53 oncogenic gain-of-functions.

The expression of miRNAs is anti-correlated with that of their target mRNAs. Therefore, we carried out an integrated miRNA-mRNA expression analysis to identify potential miRNA-mRNA modules through which GOF mutant p53 could impart its oncogenic functions (Figure 3). We determined the anti-correlated miRNA-mRNA modules by integrated analysis of miRNA and mRNA expression in H1299 cells expressing mutant p53^R273H^. For this we first analyzed the transcriptome data in H1299 cells with or without R273H mutant p53 [39]. Next, using the miRNA-mRNA expression pairing tool in IPA, we identified anti-correlated miRNA-target mRNA pairs in H1299 cells expressing mutant p53^R273H^ (Figure 3). Twelve up-regulated and eight down-regulated miRNAs showed an inverse correlation with the expression patterns of their target genes in presence of mutant p53^R273H^ (Figure 3 and Appendix A). Genes, like *TGFB1*, *SERPINA1*, *FOSL1*, and *SRGAP1*, were targeted by more than one miRNA, suggesting their regulation by multiple miRNAs in mutant p53 cancer cells. 

Notably, the analyses showed that most of the target genes of mutant p53^R273H^-regulated miRNAs were down-regulated in the mutant p53 cells, which thereby suggests an important role of cellular miRNAs in mediating GOF mutant p53-mediated regulation of cellular gene expression. 

### 3.2. Validation of GOF Mutant p53-Regulated MiRNAs in Lung Adenocarcinoma (LUAD) Patients

Next, the mutant p53^R273H^-regulated miRNAs that were obtained from our sequencing study were validated in lung adenocarcinoma patients having wild-type or missense mutant p53 from the TCGA LUAD dataset (https://tcga-data.nci.nih.gov/docs/publications/luad_2014/). Seven miRNAs were found to be significantly (*p*-value < 0.05) altered in patients with p53 mutations and showed expression patterns that were similar to that observed in our cell line experiments (Figure 4A). Two miRNAs (miR-132 and -147b) were significantly up-regulated, whereas five (miR-99a, -218, -30d, -24, and -625) were found to be significantly down-regulated (Figure 4A,B). The association observed between the expression patterns of these miRNA with p53 mutation status in lung cancer patients might point towards their potential role in mediating mutant p53-driven cancer phenotypes. It is notable, however, that only seven of the mutant p53^R273H^ regulated miRNAs could be successfully validated in TCGA patient dataset. This might be attributed to genetic heterogeneity as well as cell type compositions amongst the patients, a phenomenon that is not generally observed in cell lines. Additionally, the differential effects of various p53 mutants upon miRNAs or target genes cannot be overruled.

The association of p53 mutations with tumor aggressiveness and metastasis, including that in NSCLC, is well demonstrated [15,40,41]. Therefore, we sought to determine the possible contribution of these seven mutant p53^R273H^-regulated miRNAs in predicting lymph node metastasis (LNM) in lung adenocarcinoma patients. To this end, the relative expression levels of individual miRNAs were compared between lymph node positive (N_+_) and lymph node negative (N_0_) group of patients (Figure 5A and Appendix A). The up-regulated miRNAs (miR-132 and miR-147b) showed significantly (*p*-value ≤ 0.05) higher expression in N_+_ patients when compared to the N_0_ group (Figure 5A), thereby suggesting a possible role of these miRNAs in driving lymph node metastasis in lung cancer.

Among the five down-regulated miRNAs, the expression of only miR-30d was found to be significantly (*p*-value ≤ 0.05) low in N_+_ patients when compared to those without lymph node involvement (Figure 5A). The observation suggests a negative role of miR-30d in lymphatic metastasis during lung cancer progression. We validated these three miRNAs (miR-132, miR-147b, and miR-30d) in predicting lung cancer lymph node metastasis while using ROC (Receiver operating characteristic) curve analysis to further strengthen these results. For all three miRNAs, the LNM prediction accuracy was found to be statistically significant (*p*-value ≤ 0.05) with an area under the curve (AUC) > 0.5 (95% CI) (Figure 5B), thereby suggesting these miRNAs as significant predictors of LNM in lung adenocarcinoma. To further investigate the prognostic significance of mutant p53-regulated miRNAs in lung adenocarcinoma, we assessed their correlation with patients’ overall survival using Kaplan–Meier survival analysis. Among the seven miRNAs, expression of miR-132 and miR-99a were found to be significantly (Log-rank *p*-value ≤ 0.05) associated with patients’ survival (Figure 6 and Appendix A). The analysis showed poor overall survival in patients with high miR-132 expression (Median survival, 35.52 months; Hazard ratio, 2.679), as well as in those with low miR-99a expression (Median survival, 32.82 months; Hazard ratio, 2.113). Taken together, our analyses suggest a significant contribution of mutant p53-regulated miRNAs in determining the clinical outcome in lung adenocarcinoma patients. 

### 3.3. Mutant p53-Regulated MiR-194 and MiR-378a Predict EMT Phenotype in Cancer Cells 

GOF mutant p53 promotes EMT by modulating the expression of genes that are involved in cell adhesion, invasion, and migration [42,43,44]. Moreover, miRNAs are implicated in mutant p53-driven EMT process [26,45]. Hence, we investigated whether mutant p53^R273H^-regulated miRNAs that were obtained in our study could also predict EMT in cancer cells. To this end, we conducted an integrated analysis using the NCI-60 miRNA expression dataset [46]. Among the 47 differentially regulated miRNAs identified in the present study, information on 37 were available in the NCI-60 dataset (GSE26375). Based on the E-Cadherin/Vimentin expression ratio [36], the NCI-60 cell lines were first divided into epithelial and mesenchymal cell types and differential expression analysis for the 37 miRNAs was subsequently carried out between these two groups. Two miRNAs, miR-194 and miR-378a, were found to be significantly (*p*-value ≤ 0.05) down-regulated in the mesenchymal group as compared to that of epithelial cells (Figure 7). Previous reports also suggested the involvement of these two miRNAs in EMT and cancer metastasis [47,48]. Furthermore, negative regulation of miR-194 by mutant p53 has also been shown in endometrial cancer cells [26]. In view of these previous reports, the results of our analyses point towards an important role of mir-194 and miR-378a in mediating the GOF mutant p53-driven EMT process in cancer cells.

### 3.4. Discovery of a Novel MiRNA

Next-generation deep sequencing coupled with high throughput bioinformatic analyses enables researchers to discover novel miRNAs with high sensitivity and specificity [49]. Here, while using the sequencing reads that did not align to the miRbase (miRBase20; http://www.mirbase.org/), we tried to identify candidate sequences that have potential precursor miRNA hairpin structures around their genomic coordinates. We identified three putative hairpin precursor miRNAs. One of them is in the intergenic region of chromosome 1 (chr1:168239822-168239840), while the other two overlaps with the tRNA-Ile-ATT (chr6:27,636,363-27,636,381) and tRNA-Asp-GAY (chr12:96,429,800-96,429,817). These two candidates might have been falsely determined and they could be part of the degraded tRNA. However tRNA-derived RNA fragments [50] and miRNAs [51] are also being reported in the literature. The sequence that was mapped to the intergenic region of the genome was considered for further validation. The secondary structure of its precursor miRNA showed a characteristic stem-loop structure (Figure 8A) with ΔG = −27.40 kcal/mol and the mature sequence mainly lies on the stem region. Interestingly, the novel miRNA, designated as “miR-X” was found to be homologous to the efu-miR-9277 of bat, which suggests its possible existence. Furthermore, we have used the publicly available dataset (GSE33858) of lung adenocarcinoma, squamous cell carcinoma, and corresponding noncancerous tissue samples, and evaluated the expression profile of miR-X. Integrative Genomics Viewer of the primary aligned reads across the miR-X region showed the expression of miR-X in some of lung adenocarcinoma, squamous cell carcinoma, and adjacent noncancerous tissues (Appendix A). miR-X was found to be upregulated in some of the adenocarcinoma and squamous cell carcinoma tissues as compared to the corresponding adjacent noncancerous tissues of the same patients but was not significant at the level of *p* = 0.05 (Figure 8B). To explore the biological functions of miR-X, we predicted its downstream targets while using miRanda [31] and identified several putative target genes (Appendix A). Subsequent pathway analysis of the predicted target genes showed significant enrichment of important biological pathways (Appendix A). Moreover, network analyses of the target genes indicated the possible involvement of miR-X in crucial cellular processes that are commonly deregulated in cancer, including cell growth and proliferation; cellular assembly and organization, DNA replication, recombination and repair, and cell-to-cell signaling (Appendix A).

### 3.5. Novel miR-X Promotes Oncogenic Properties in Lung Adenocarcinoma Cells

Next, we aimed to investigate the expression levels of the newly identified miRNA in cancer cells harboring GOF mutant p53. To this aim, we compared the relative expression of miR-X between H1299/EV and H1299/mutant p53^R273H^ stable cell lines while using qRT-PCR. We found significant up-regulation of miR-X in H1299 cells harboring mutant p53^R273H^ when compared to the control cells (Figure 9A). To further asses its possible biological functions, we investigated the cellular effect of miR-X overexpression in different cell-based assays. For this, we cloned the newly identified miRNA into pRNA.U6 miRNA expression vector and subsequently overexpressed in H1299 cells (Figure 9B). The ectopic expression of miR-X led to a marked increase in the proliferation rate of H1299 cells, as measured by WST-1 cell proliferation assay (Figure 9C), which was one of the most significantly enriched networks for the miR-X target genes (Appendix A). We overexpressed miR-X in another lung cancer cell line A549 to further validate this observation. A549 cells overexpressing miR-X also showed significantly enhanced proliferation as compared to the control empty vector transfected cells (Figure 9C). Consistent with these results, we observed significantly higher percentage of Ki67 positive cells in both H1299 and A549 cells upon miR-X overexpression, thereby confirming its role in promoting lung cancer cell proliferation (Figure 9D,E). The growth promoting effect of miR-X was further recapitulated in subsequent colony formation assay, where ectopic expression of the miRNA clearly promoted the colony forming ability in H1299 cells (Figure 9F). We performed a monolayer wound healing assay to determine whether miR-X could also promote cancer cells migration. When compared to the empty vector-transfected H1299 cells, cells overexpressing miR-X was found to migrate significantly faster, as measured by the relative distance that is covered by the cells after the scratch was introduced (Figure 9G). Together, these results suggest a potential role of miR-X in driving oncogenic phenotypes. 

## 4. Discussion 

GOF mutant p53, an oncogenic transcription factor, transactivates or represses a diverse set of genes to promote tumorigenesis [3,5,9,35,52,53]. Beside this, the regulation of cellular miRNAs is also implicated in different oncogenic phenotypes conferred by mutant p53 [24,25,26,27,28,29]. Therefore, identification of miRNAs regulated by mutant p53 on a global scale would provide further insights into the cellular pathways and biological functions that are governed by GOF mutant p53 in the context of tumorigenesis. In this study, we explored GOF mutant p53^R273H^-regulated miRnome in lung adenocarcinoma cells while using small RNA deep sequencing. Subsequent target gene prediction and network enrichment analyses revealed that mutant p53^R273H^-regulated miRNAs interact with several crucial genes and molecular pathways that are implicated in tumorigenesis. Enrichment of p53 in the miRNA regulatory network indicates that specific miRNAs (miR-99a, -132, -140, -17, -194, and -296) are most likely to be regulated by both wild-type and mutant forms of p53, as also suggested by previous studies [23,54,55,56]. Our analyses also predicted molecular interactions between mutant p53-regulated miRNAs and cancer metastasis-associated genes *TGFBI*, *EPHB6,* and insulin signaling, a pathway that is commonly deregulated in cancer [57,58]. Integrated analysis of miRNA and mRNA expression profiles further identified potential miRNA-mRNA modules that might be critical in mediating mutant p53 gain-of-functions in cancer cells. 

We validated our cell-based data in the TCGA lung adenocarcinoma patient dataset to further investigate the prognostic value of the mutant p53^R273H^-regulated miRNAs. Approximately 30% of patients in the TCGA dataset had missense mutations in *TP53*, whereas 16% carried the truncated version of the protein. High *TP53* mutation frequencies in lung cancer patients clearly indicate the importance of studying the mutant p53-specific regulation of miRNAs. Eighteen miRNAs that were detected in our cell line study showed similar expression patterns in the patient samples, however, only seven of them were statistically significant. It is quite reasonable to consider that different mutant p53 proteins might differentially regulate miRNAs that were obtained from our cell line study since the dataset contains patients with different types of *TP53* mutation. Genetic heterogeneity among patients might also account for the limited agreement of our cell line data with the TCGA dataset. Several studies evidenced the association of tumor metastasis with *TP53* mutations in different types of cancers [40,41]. Here, we investigated the possible contribution of mutant p53^R273H^-regulated miRNAs in predicting lymph node metastasis in lung adenocarcinoma patients by systematic bioinformatics analyses of the TCGA LUAD dataset. Among the seven mutant p53^R273H^-regulated miRNAs, the expression of miR-132, -147b, and -30d was found to be significantly correlated with the LNM status of the patients. ROC curve analysis further confirmed that these miRNAs are significant predictors of LNM in NSCLC. We hypothesize that mutant p53 modulates the expression of these miRNAs to drive lymph node metastasis in NSCLC, as *TP53* mutations generally precede node metastasis in lung cancer [15]. However, further studies are required to ascertain the contribution of these miRNAs in node metastasis. This has been suggested that miRNAs are potential contributors to mutant p53-driven EMT phenotype [26]. In agreement with this, our integrated analyses of the NCI-60 miRNA expression dataset identified mutant p53^R273H^-regulated miR-194 and miR-378a as potential determinants of EMT in cancer cell lines. Taken together, our analyses clearly indicate an important role of cellular miRNAs in mediating diverse oncogenic properties of GOF mutant p53 in human cancer. 

It is anticipated that many more miRNAs are yet to be discovered than presently annotated in public repositories [59]. In this study, we discovered a novel miRNA from the unmapped sequence reads of the genome-wide small RNA sequencing data. Further analyses using publicly available small RNA sequencing dataset of lung cancer tissues showed enough primary reads to be precisely aligned to the genomic region of the newly identified miRNA. This suggests that miR-X, although not identified earlier, was present in lung cancer and adjacent benign lung tissues. The precursor miR-X overlaps with two LTRs of ERVL family (HERVL18-int and LTR18A), while 13 bases of the mature miR-X overlap with one of the LTRs and remaining five bases are in the non-LTR region. However, it should be noted that ~20% of the human miRNAs are reported to have complete or partial overlap with transposable elements, and many of them are proposed to be derived from transposable elements, including LTR, LINE, and SINE [60,61,62]. Target prediction and pathway analyses revealed that the novel miRNA targets a wide array of genes that are involved in different signaling pathways. Subsequent cell-based assays further indicated that the newly identified miRNA could promote oncogenic phenotypes, including cancer cell proliferation, clonogenicity, and migration, which suggests its potential role in oncogenesis. Increased expression of the novel miRNA in GOF mutant p53 expressing cancer cells further suggests a plausible role of this miRNA in mediating mutant p53-driven tumorigenesis. However, we could not see any significant difference in miR-X expression between adjacent normal and lung tumor tissues, at least in the publicly available small RNA sequencing dataset that we analyzed in the present study. This might be attributed to the tumor heterogeneity or insufficient number of tumor samples analyzed. Although further functional studies are required to fully characterize the novel miRNA, it holds the potential to be included in the growing list of cellular miRNAs. Collectively, our study provides an important framework for further research on understanding the molecular mechanism of GOF mutant p53-driven tumorigenesis and identified a specific miRNA signature that might be useful in predicting NSCLC prognosis.

## Figures and Tables

**Figure 1 genes-10-00852-f001:**
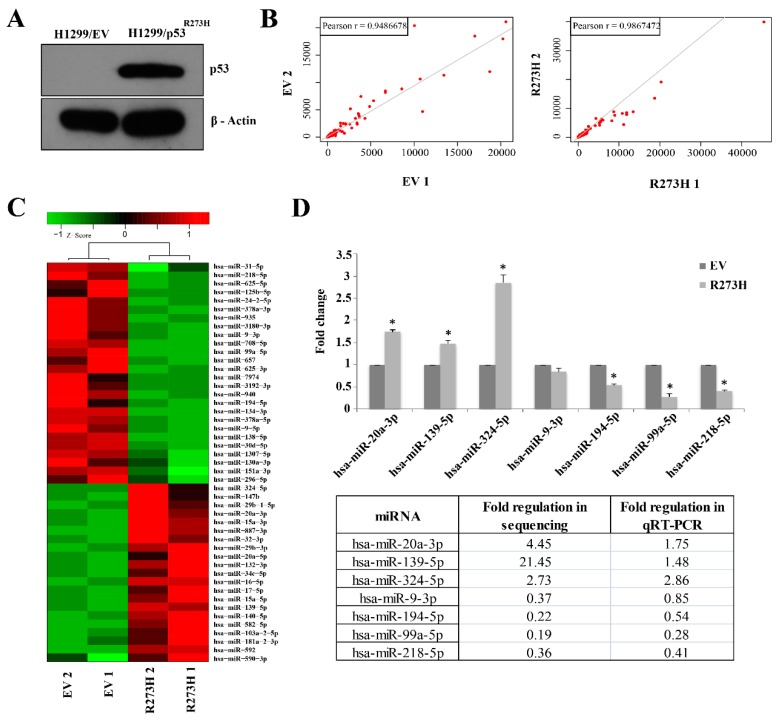
miRNA expression profiling in H1299 cells expressing mutant p53^R273H^ using small RNA sequencing. (**A**) Immunoblot showing mutant p53^R273H^ level in H1299/ mutant p53^R273H^ stable cells. (**B**) Scatter plots showing a correlation of normalized read counts between biological replicates of individual samples. (**C**) Heat map showing normalized read counts of miRNAs differentially expressed (*p*-value ≤ 0.05) between H1299/mutant p53^R273H^ and H1299/EV cells. Hierarchical clustering of samples is shown. Color bar indicates Z- scores of normalized read counts. Red color indicates high expression, green color indicates low expression. (**D**) Validation of the selected differentially expressed miRNAs in H1299/mutant p53^R273H^ cells using qRT-PCR. Bar graphs represent mean ± s.d.; *n*≥ 2; two-tailed Student’s *t*-test: * *p* < 0.05. A relative comparison of qRT-PCR data with the deep sequencing results is shown.

**Figure 2 genes-10-00852-f002:**
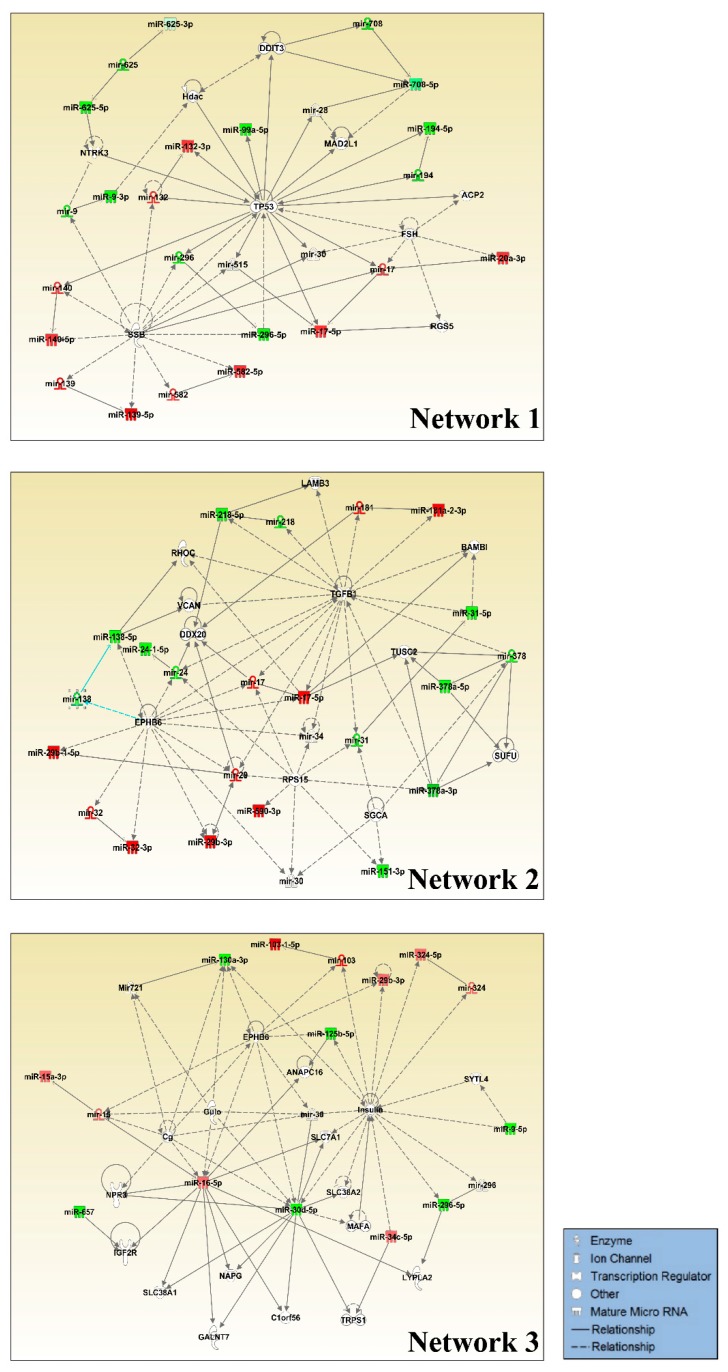
Molecular networks enriched by miRNAs differentially expressed in H1299/mutant p53^R273H^ cells. Ingenuity Pathway Analysis (IPA) generated top three significantly enriched regulatory networks of mutant p53^R273H^ regulated miRNAs. The networks illustrate direct or indirect interactions between altered miRNAs and their target genes. Only the highlighted (green/red) miRNAs were present in our dataset. Green and red represent down-regulated and up-regulated miRNAs, respectively.

**Figure 3 genes-10-00852-f003:**
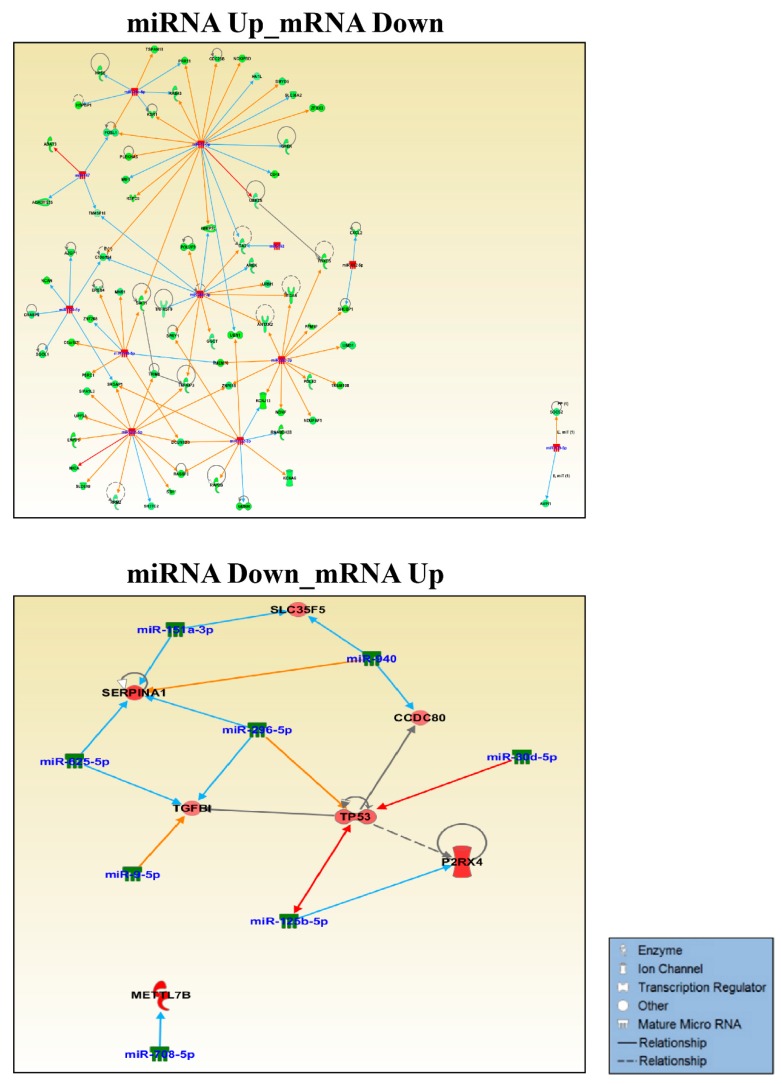
Molecular networks of anti-correlated miRNA-mRNA pairs in H1299/mutant p53^R273H^ cells derived from the integrated miRNA-mRNA expression analysis. Target genes of the up-regulated miRNAs (Red) were down-regulated (Green) [upper panel] and those of down-regulated miRNAs (Green) were up-regulated (Red) [lower panel] in the presence of mutant p53^R273H^. The color intensity of the nodes indicates relative miRNA or mRNA expression levels in presence of mutant p53^R273H^. The types of interaction between miRNAs and their respective target mRNAs are represented by arrows in dark red (experimentally validated), bright orange (highly predicted), and light blue (moderately predicted).

**Figure 4 genes-10-00852-f004:**
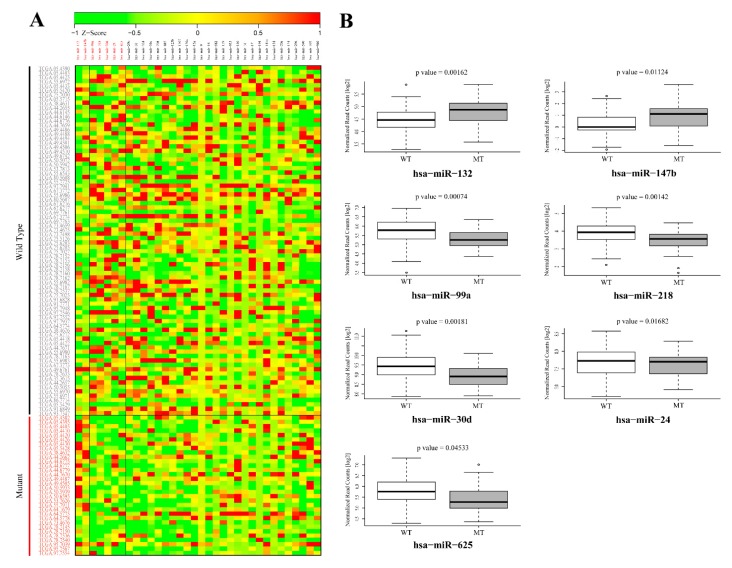
Validation of mutant p53^R273H^-regulated miRNAs in the TCGA lung adenocarcinoma patient dataset. (**A**) Heat map showing normalized read counts of mutant p53^R273H^-regulated miRNAs across TCGA lung adenocarcinoma patients bearing wild type and mutant p53. Based on *TP53* mutation status, TCGA patients were categorized into two groups, wild type and mutant p53 patients. The relative expression levels of mutant p53-regulated miRNAs obtained in the present study were subsequently validated by comparing their normalized read counts between these two groups of patients. Patients with wild type p53 and mutant p53 are shown in black and red color letters respectively. miRNAs shown in red are significantly (*p* < 0.05) altered between wild-type and mutant p53 patients with a pattern similar to that of our small RNA sequencing results. Heat map scale bar indicates Z-scores of normalized read count values. (**B**) Box-Whisker plots showing log2 transformed normalized read counts of two up-regulated and five down-regulated miRNAs in patients with mutant p53. *p*-values are indicated.

**Figure 5 genes-10-00852-f005:**
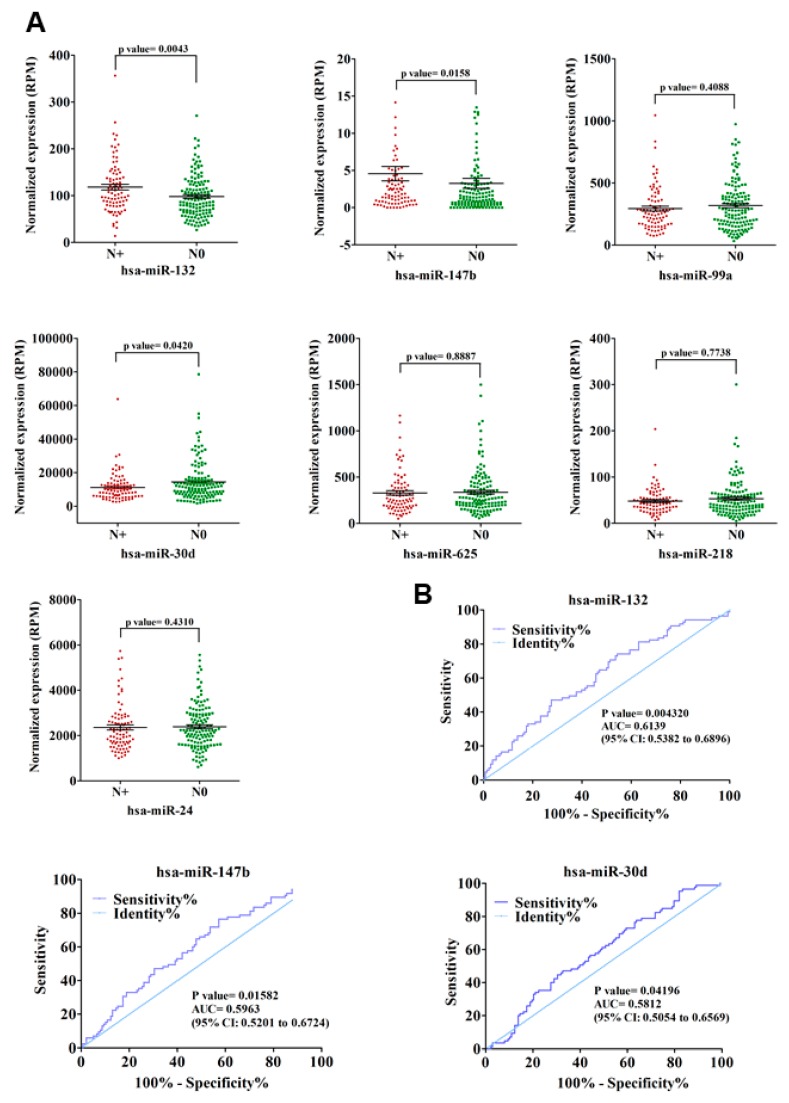
Mutant p53^R273H^-regulated miRNAs predict lymph node metastasis (LNM) in lung adenocarcinoma patients. (**A**) Relative expression of seven mutant p53-regulated miRNAs in lymph node positive (N_+_, *n* = 85) and lymph node negative (N_0_, *n* = 138) patients. Scatter plots showing normalized expression (RPM) of individual miRNAs in N_+_ and N_0_ group of patients. *p*-values are indicated. (**B**) Receiver operating characteristic (ROC) curve analyses for LNM prediction in lung adenocarcinoma patients using relative expression of miR-132, miR-147b, and miR-30d. Area under the receiver operating characteristic curve (AUC) is shown. p -values are indicated.

**Figure 6 genes-10-00852-f006:**
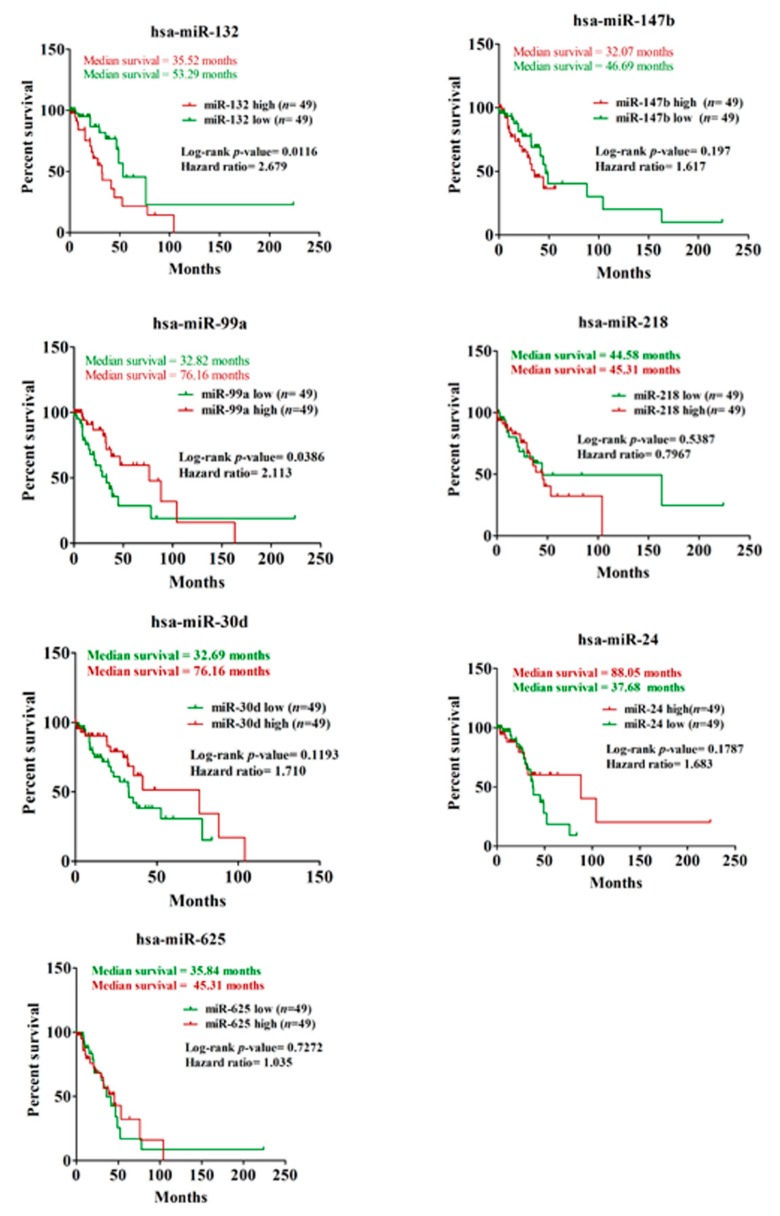
Mutant p53^R273H^-regulated miRNAs determine poor survival in lung adenocarcinoma patients. Kaplan–Meier analyses showing relative survival probabilities of lung adenocarcinoma patients with high (≥75th percentile) and low (≤25th percentile) expression of the individual mutant p53-regulated miRNAs. Log-rank *p*-value, hazard ratio, and median survival time are indicated. *n* indicates the number of patients in respective miRNA low and high groups.

**Figure 7 genes-10-00852-f007:**
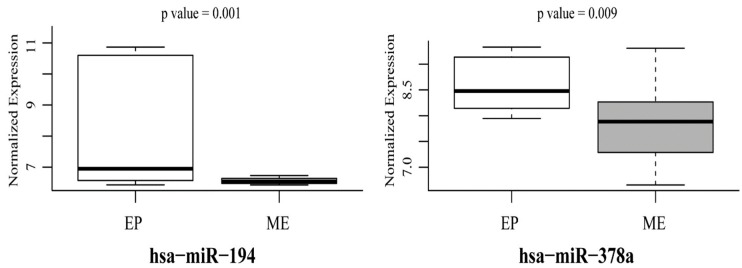
GOF Mutant p53^R273H^-regulated miR-194 and miR-378a is down-regulated in NCI-60 cell lines with mesenchymal phenotype. Box-Whisker plots showing relative expression of miR-194 and miR-378a in mesenchymal and epithelial groups of NCI-60 cell lines. *p*-values are indicated.

**Figure 8 genes-10-00852-f008:**
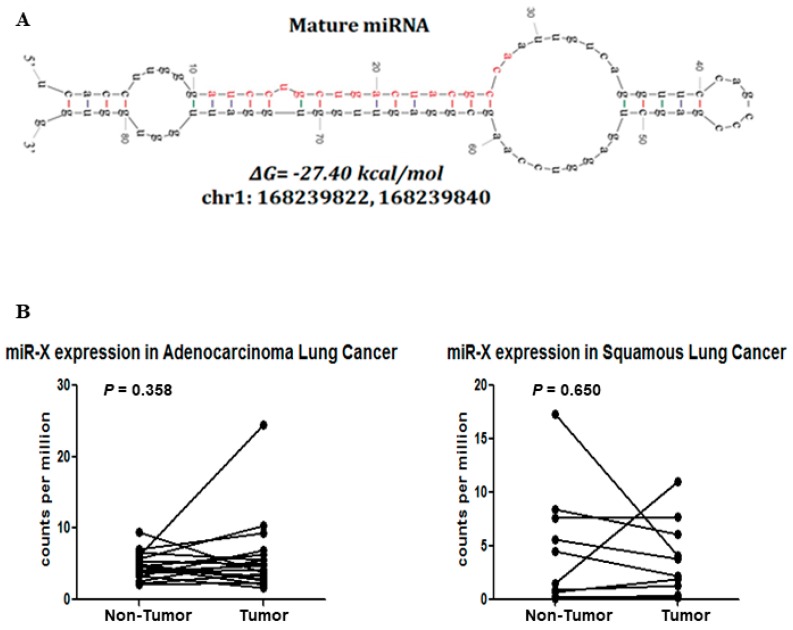
Discovery of the novel miRNA. (**A**) Stem-loop structure of the novel miRNA (miR-X) precursor predicted by mfold version 3.6. The mature miRNA sequence is shown in red. The minimum free energy (MFE; ΔG) of hairpin structure and the genomic coordinate of the mature sequence are indicated. (**B**) Symbols and line graphs showing relative expression (counts per million) levels of miR-X in paired tumor and non-tumor lung adenocarcinoma (left panel) or squamous cell carcinoma (right panel) tissues. *p*-values are indicated.

**Figure 9 genes-10-00852-f009:**
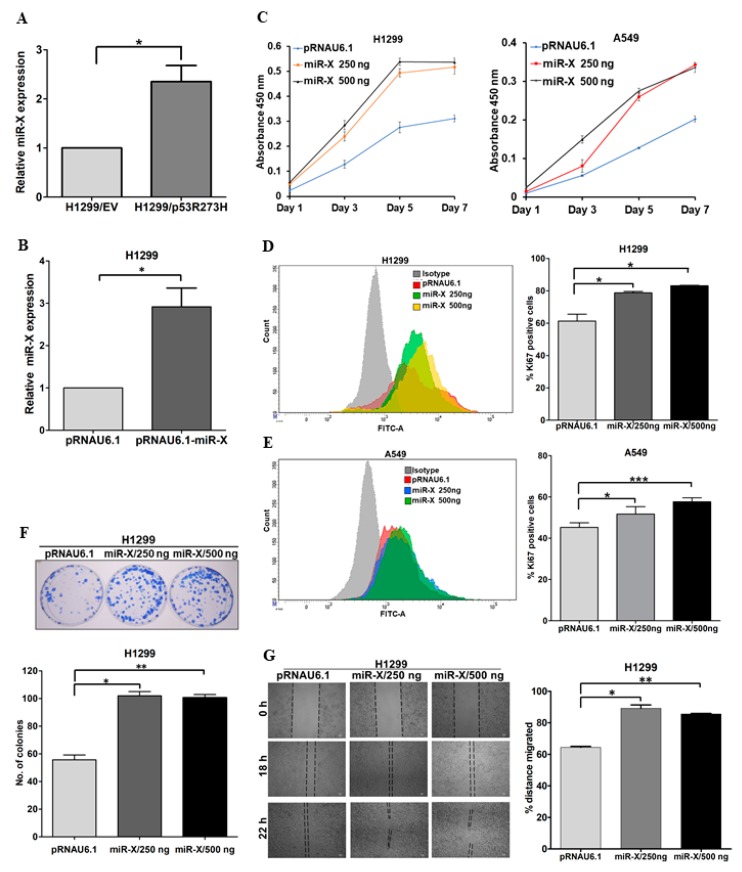
miR-X promotes oncogenic properties in lung cancer cells. (**A**) QRT-PCR data showing relative expression levels of miR-X in H1299/EV and H1299/mutant p53^R273H^ stable cell lines (**B**) QRT-PCR data showing ectopic expression of miR-X in H1299 cells transfected with pRNAU6.1-miR-X. H1299 cells were transfected with either empty pRNAU6.1 vector or with 250 ng of pRNAU6.1-miR-X. Forty eight hours post-transfection relative miR-X expression was evaluated using qRT-PCR. (**C**) Bar graphs showing relative proliferation of H1299 (left panel) and A549 (right panel) cells upon ectopic expression of miR-X as measured by WST-1 cell proliferation assay. Cells were transfected with either empty vector or miR-X expression plasmid and 16 h post-transfection WST-1 assay was performed for the day 1 and for the days indicated. Data represents average absorbance values at 450 nm of three independent experiments. (**D**,**E**) (Left panels) Histogram showing mean fluorescent intensities of empty vector and miR-X transfected H1299 (D) and A549 (E) cells stained with anti-Ki-67 antibody as measured by FACS analyses. (Right panels) Bar graphs showing relative percentages of Ki-67 positive H1299 (D) and A549 (E) cells transfected with control empty vector or with miR-X expression plasmid. (**F**) (Upper panel) Representative images of colonies formed by H1299 cells upon miR-X overexpression in clonogenic assay. (Lower panel) Quantification of the data that were obtained from the clonogenic assay. (**G**) (Left panel) Representative images of wound healing assay performed in H1299 cells transfected with either empty vector or miR-X expression plasmid. Images were captured at 0 h, 18 h, and 22 h after the scratch was introduced. (Right panel) Bar graph showing relative percentage of distance covered by the cells through 18 h post-scratch. Data represent mean ± s.d.; *n* = 3; two-tailed Student’s *t*-test: * *p* <0.05, ** *p* <0.01, *** *p* <0.001.

**Table 1 genes-10-00852-t001:** Sequencing summary of the small RNA libraries. Raw reads were quality processed and aligned to the reference human genome (GRCh37/hg19) and miRBase (miRBase20). Total number and the percentage of processed reads mapped to miRNA, tRNA, rRNA, and adaptors are shown. The number of miRNAs identified in each sample with ≥1 or ≥3 read counts are indicated.

Samples		H1299/EV Replicate 1	H1299/EV Replicate 2	H1299/R273H Replicate 1	H1299/R273H Replicate 2
**Total Raw Reads**		1,442,378	751,538	1,292,998	874,753
Reads after pre-processing		753,866	340,244	566,961	500,523
Mapping to References (miRBase, rRNA, tRNA & Adaptor)	Total Reads	582,481	261,368	462,283	420,018
	Percentage	77.27	76.82	81.54	83.92
Reads mapped to miRNA	Total Reads	540,145	251,022	453,934	408,055
	Percentage	71.65	73.78	80.06	81.53
Reads mapped to tRNA	Total Reads	40,071	9763	7932	11,191
	Percentage	5.32	2.87	1.40	2.24
Reads mapped to rRNA	Total Reads	2235	575	406	767
	Percentage	0.30	0.17	0.07	0.15
Reads mapped to adaptor	Total Reads	30	8	11	5
	Percentage	0.0040	0.0024	0.0019	0.0010
Total expressed miRNA (1 or more reads)		692	512	507	518
miRNA (3 or more reads)		363	245	299	306

**Table 2 genes-10-00852-t002:** Molecular and cellular functions enriched in H1299/mutant p53^R273H^ cells.

H1299/EV vs H1299/mutant p53^R273H^		
Molecular and Cellular Functions		
Name	*p*-Value	# Molecules
Cell Cycle	7.36E-06-4.52E-02	8
Cellular Development	7.50E-05-4.82E-02	13
Cellular Growth and Proliferation	7.50E-05-4.82E-02	12
Cellular Movement	2.72E-04-3.14E-02	10
DNA Replication, Recombination and Repair	3.67E-04-4.82E-02	5

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
