# Peer review of "Genome-Wide Small RNA Sequencing Identifies MicroRNAs Deregulated in Non-Small Cell Lung Carcinoma Harboring Gain-of-Function Mutant p53"

_genes, 2019, doi:10.3390/genes10110852_

Round 1
Reviewer 1 Report
In the manuscript entitled „ Genome-wide Small RNA Sequencing Identifies microRNAs Deregulated in Non-Small Cell Lung Carcinoma Harboring Gain-of- Function Mutant p53“ the authors described the influence of P53 mutation on miRNA expression. They discovered a until now unknown miRNA with oncogenic potential.
This is a very interesting manuscript, which should be published.
A few points need correction prior to publication.
In line 277 pathway analysis is mentioned. Please indicate the system, which was used for this pathway analysis. In addition, from my point of view, at least the most important part of Figure S1 should be shown within the manuscript. Fig. 1E,F need more explanation and should be shown in a way (size) that enables identification of the interaction partners. Table 1 please explain the terms U1,U2 Fig. 2A, please explain this figure more clearly so that a reader can retrace your statement of lines 320-324. Please explain the n=49 of Fig. 4 Fig. 5: How were mesenchymal and epithelial groups of NCI-60 cell lines distinguished. Please give details of assignment with miRNAs. line 399, please explain the term unmapped sequence read. Fig. 7 please describe all pictures in a readable form (size) Spelling errors should be corrected.
Author Response
Response to the Reviewers
We thank both the reviewers for valuable comments which have substantially improved the quality of the manuscript. A point by point reply to the reviewers’ suggestions are given below:
Response to Reviewer 1 Comments
Point 1: In line 277 pathway analysis is mentioned. Please indicate the system, which was used for this pathway analysis. In addition, from my point of view, at least the most important part of Figure S1 should be shown within the manuscript.
Response 1: We have used Ingenuity Pathways Analysis tool (IPA; https://analysis.ingenuity.com) for the pathway analyses. This has been mentioned in the Materials and Methods section (Line 180 of the revised manuscript) as well as in the revised Results section (Line 329 of the revised manuscript text).
As suggested, we have now incorporated a table describing molecular and cellular functions altered by miRNAs in H1299/mutant p53R273H bearing cells within the manuscript text. Please see the Table 2.
Point 2: Fig. 1E, F need more explanation and should be shown in a way (size) that enables identification of the interaction partners.
Response 2: As suggested, Fig. 1E and 1F is now further explained within the Results section of the revised manuscript (Line 334-366 of the revised manuscript text).
We have now increased the overall size and resolution of the images so that the name of the interactors is legible. Fig. 1E and 1F are now provided as Fig. 2 and Fig. 3 respectively in the revised manuscript.
Point 3: Table 1 please explain the terms U1, U2.
Response 3: U1 and U2 were originally given to indicate biological replicates unit 1 and 2 respectively. For clarity, we have now relabeled them as “Replicate 1” and “Replicate 2” in the revised manuscript.
Point 4: Fig. 2A, please explain this figure more clearly so that a reader can retrace your statement of lines 320-324. Please explain the n=49 of Fig. 4.
Response 4: As suggested, we have now explained Fig. 2A more clearly in the respective figure legends (Figure 4A of the revised manuscript).
In the original paper, in Figure 4, n=49 means the number of patients analyzed. This has now been clearly mentioned in Figure 6 legend of the revised manuscript.
Point 5: Fig. 5: How were mesenchymal and epithelial groups of NCI-60 cell lines distinguished. Please give details of assignment with miRNAs.
Response 5: The NCI-60 cell lines were classified into mesenchymal and epithelial groups based on the E-Cadherin/Vimentin expression ratio. The miRNA expression values of these cell lines were obtained from the NCI-60 expression dataset. These expression values were subsequently compared between these two groups of NCI-60 cell lines. We have explained this analytical strategy in the Materials and Methods section (Line 226-231) of the revised manuscript as well as in the revised Results section (Line 637-643).
Point 6: line 399, please explain the term unmapped sequence read.
Response 6: We have removed the term “unmapped sequence read” and revised the text in the manuscript (Line 668-669 of the revised manuscript text).
Point 7: Fig. 7 please describe all pictures in a readable form (size). Spelling errors should be corrected.
Response 7: As suggested, we have now reformatted Fig.7 (Figure 9 of the revised manuscript) and increased the font size in the individual figure panels.
Spelling errors have been corrected.
Reviewer 2 Report
Manuscript ID: genes-591077
In this paper, the authors identified a novel miRNA and named ‘miR-X’ in mutant p53R273H bearing non-small cell lung carcinoma (NSCLC) cells using small RNA deep sequencing. The newly identified miRNA promotes proliferation, colony-forming ability and migration of NSCLC cells. This is a potentially interesting paper. However, still several issues need to be answered and clarified.
More than 1,000 human microRNAs are identified and registered in miRNA database (e.g. miRBase). Why is this miR-X unidentified so far? The authors should discuss this issue. There is no data presented in this study concerning the expression levels of miR-X in normal and cancer human cells. The authors should validate the sequencing reads mapped to the miR-X using published databases of small RNA sequencing. The authors can also validate the expression level of miR-X by real-time PCR analysis (e.g. TaqMan PCR) in normal and cancer cells. The authors should address the correlation between miR-X expression levels and p53 mutational status in human cancer cells.Author Response
Response to the Reviewers
We thank both the reviewers for valuable comments which have substantially improved the quality of the manuscript. A point by point reply to the reviewers’ suggestions are given below:
Response to Reviewer 2 Comments
Point 1: More than 1,000 human microRNAs are identified and registered in the miRNA database (e.g. miRBase). Why is this miR-X unidentified so far? The authors should discuss this issue. There is no data presented in this study concerning the expression levels of miR-X in normal and cancer human cells. The authors should validate the sequencing reads mapped to the miR-X using published databases of small RNA sequencing. The authors can also validate the expression level of miR-X by real-time PCR analysis (e.g. TaqMan PCR) in normal and cancer cells.
Response 1: We are thankful to the reviewer for raising this issue. We have now used published small RNA-seq data (BAM files) from the GEO database (GSE33858) of lung cancer tissues and adjacent non-tumor lung tissues and evaluated the expression profile of miR-X. Although we could not find any significant differential expression of miR-X, however, a sufficient number of primary reads are found to be aligned, precisely in this region. This suggests that miR-X, although not discovered earlier, but was present in lung cancer and adjacent benign lung tissues. We have also included the expression profile of miR-X both in Lung Adenocarcinoma and Squamous cell carcinoma samples compared to the adjacent (benign) normal lung tissues, obtained from the GEO database (GSE33858). In the revised manuscript, we have now included a figure with the relative expression of miR-X (Figure 8B) and included the IGV screenshot (Supplementary Figure S3) of the aligned reads across miR-X. The results have been discussed within the Results (Line 680-687) as well as in the Discussion section (Line 909-920) of the revised manuscript.
Point 2: The authors should address the correlation between miR-X expression levels and p53 mutational status in human cancer cells.
Response 2: We have now tested the relative expression levels of miR-X in H1299 cells expressing mutant p53R273H using qRT-PCR. The results (Figure 9A) have been discussed within the Result section (Line 711-715) of the revised manuscript text.
Round 2
Reviewer 2 Report
Previous concerns have been adequately addressed.
Author Response
We thank the reviewer for the valuable suggestions which significantly improved the overall quality of the manuscript.